# Ancient hydrothermal seafloor deposits in Eridania basin on Mars

Joseph R. Michalski[1], Eldar Z. Noe Dobrea[2], Paul B. Niles[3] & Javier Cuadros[4]

The Eridania region in the southern highlands of Mars once contained a vast inland sea with a volume of water greater than that of all other Martian lakes combined. Here we show that the most ancient materials within Eridania are thick ( > 400 m), massive (not bedded), mottled deposits containing saponite, talc-saponite, Fe-rich mica (for example, glauconite-non-tronite), Fe- and Mg-serpentine, Mg-Fe-Ca-carbonate and probable Fe-sulphide that likely formed in a deep water (500–1,500 m) hydrothermal setting. The Eridania basin occurs within some of the most ancient terrain on Mars where striking evidence for remnant magnetism might suggest an early phase of crustal spreading. The relatively well-preserved seafloor hydrothermal deposits in Eridania are contemporaneous with the earliest evidence for life on Earth in potentially similar environments 3.8 billion years ago, and might provide an invaluable window into the environmental conditions of early Earth.

[1] Department of Earth Sciences and Laboratory for Space Research, University of Hong Kong, Pokfulam Road, Hong Kong, China. [2] Planetary Science Institute, 1700 E. fort Lowell, Tucson, Arizona 85719, USA. [3] Astromaterials Research and Exploration Science, NASA Johnson Space Center, Houston, Texas, USA. [4] Department of Earth Sciences, Natural History Museum, Cromwell Road, London SW7 5BD, UK. Correspondence and requests for materials should be addressed to J.R.M. (email: jmichal@hku.hk).

The oldest supracrustal rocks on Earth are early Eoarchean seafloor deposits ($\geq 3.7$ Ga)[1]. The presence of isotopically light carbon[2] within biogenic morphologies in these rocks indicates that life may have flourished on the early Earth in hydrothermal seafloor environments[3]. Yet progress in further understanding the actual origin of life or prebiotic chemistry from these rocks, or those of similar age, is severely challenged by the fact that they have experienced multiple generations of metamorphism, metasomatism and deformation[4]. The search for life's origins through empirical geologic evidence might require exploration beyond Earth, where younger geological activity has not overwritten critically important chemical and textural records. This journey could lead to Mars where ancient sedimentary, volcanic and hydrothermal deposits contemporaneous with the origin of life on Earth have escaped deep burial and metamorphism.

The Eridania region, located at the boundary of Terra Cimmeria and Terra Sirenum (Fig. 1), includes exposures of some of the most ancient terrain on Mars[5]. This area exhibits the strongest evidence for remnant magnetism on Mars and could be a site of ancient crustal spreading[6] (Fig. 1a). Geophysical models suggest that the area had a high thermal gradient in the Noachian[7], consistent with regional magmatism. The presence of a high-potassium anomaly[8] could be an indication of a deep mantle source for ancient volcanism in the area or widespread alteration of the crust[9,10]. Regardless of whether the remnant magnetism is truly indicative of early plate tectonics, the fact that the magnetic signature is observed is an indication that the near surface materials formed when the magnetic field of Mars was strong, and have not been buried as has occurred in other areas[5]. Therefore, the geology observed here provides insights into geological processes that operated in the earliest observable epoch of Martian history[11].

In addition to containing come of the most ancient crust on Mars, the Eridania region is important because it contains a large basin that was once filled with water. In this study, we examined the geology and mineralogy of the most ancient deposits within this basin. Using infrared spectroscopy and high-resolution imaging, we show that the Eridania basin contains a complex suite of alteration minerals that likely formed in a hydrothermal seafloor volcanic-sedimentary setting.

## Results

**Geomorphic evidence for an ancient sea in Eridania basin.** Eridania basin is composed of a series of connected, smaller, quasi-circular basins (Fig. 1), which potentially originated as very ancient impacts that were resurfaced by volcanism and erosion early in Mars' history[12,13]. The extent of the Eridania basin was previously defined as the 1,100 m elevation contour around these sub-basins[13] (Fig. 1). Irwin et al.[13] deduced that the Eridania basin was once filled to this level because it is at this elevation that the 3-km-wide Ma'adim Vallis outflow channel originates (Fig. 1c)[13]. This morphology, with a complete lack of upstream tributaries, suggests that the channel formed at full width, although a spillway at the edge of the Eridania basin at $\sim 1,100$ m elevation, a strong indication that the basin was filled with water at the Noachian/Hesperian boundary[13].

Irwin et al.[13] recognized the unusual hypsometry of the Eridania basins, noting that they have unusual concave topographic profiles. We similarly compare the topographic data of Eridania basins to data of basins elsewhere on Mars (Supplementary Fig. 1). Most similar sized basins elsewhere on Mars exhibit clear 'U-shaped' topographic profiles which arise from colluvial, fluvial and volcanic resurfacing of the basins in a subaerial setting. The concave structure of the Eridania basins is

an indication that, during the only intense period of erosive activity in mars history, these basins were protected beneath water or ice-covered water.

Previous researchers noted that Noachian valley networks also terminate at an elevation of $\sim 700$–1,100 m (refs 13,14), suggesting the existence of an ancient base level. If a water level existed between 700 and 1,100 m elevation, the basin topography implies that the parts of the lake would have been 1–1.5 km deep. The approximate size of such a body of water would have been $\sim 1.1 \times 10^6$ km$^2$, $\sim 3 \times$ larger than the largest landlocked lake or sea on Earth (Caspian Sea) (Fig. 2). In fact, even a conservative estimate of the volume of the Eridania sea exceeds the total volume of all other lakes on Mars combined (Fig. 2)[15]. Here we synthesize previous work and provide new analyses of the mineralogy, geology, and context of the most ancient deposits in Eridania basin (Fig. 1c), which we argue formed in a deep-water hydrothermal setting.

**Multiple types of colles and chaos units in Eridania basin.** Unique deposits found only at the centre (deepest part) of each basin (Fig. 1c) consist of fractured and dismembered blocks

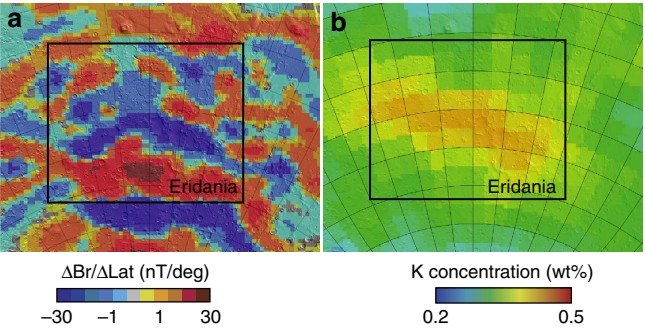

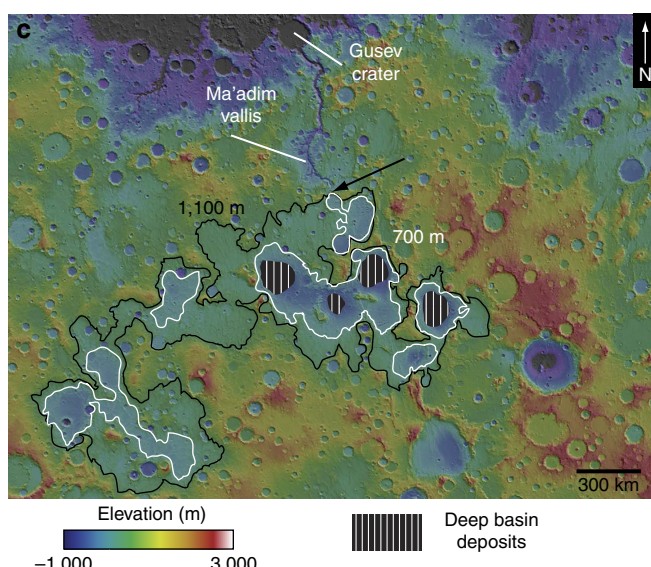

**Figure 1 | Regional context of Eridania basin.** Eridania is located at the boundary of Terrae Cimmeria and Sirenum ($\sim 180$E, 30S), an ancient part of martian crust which exhibits strong remnant magnetism (**a**) and increased abundance of potassium observed in GRS data (**b**). Eridania contains a large closed basin defined by the 1,100 m MOLA topographic contour shown here in black (**c**). This elevation marks the maximum extent of an ancient sea which spilled over to form the Ma'adim vallis channel, feeding a smaller lake in Gusev crater in the lake Noachian. A lower, perhaps more stable base level is defined by the 700 m contour shown in white (**c**). GRS, Gamma Ray Spectrometer.

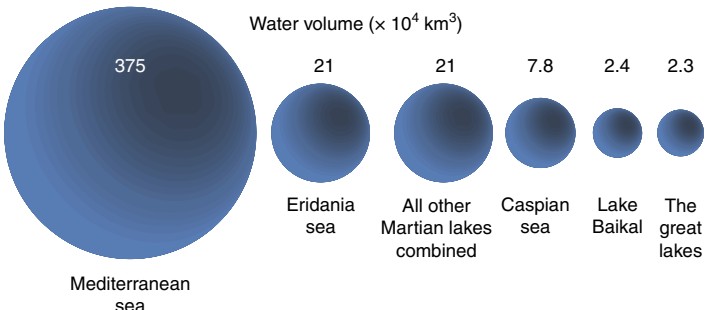

**Figure 2 | Volume of water in Eridania basin.** A scale model compares the volume of water contained in lakes and seas on the Earth and Mars to the estimated volume of water contained in an ancient Eridania sea. Even using a conservative estimate of the Eridania water volume, this amount exceeds the estimated volume of all other closed basin lakes on Mars combined[15] and far exceeds that of any terrestrial lake.

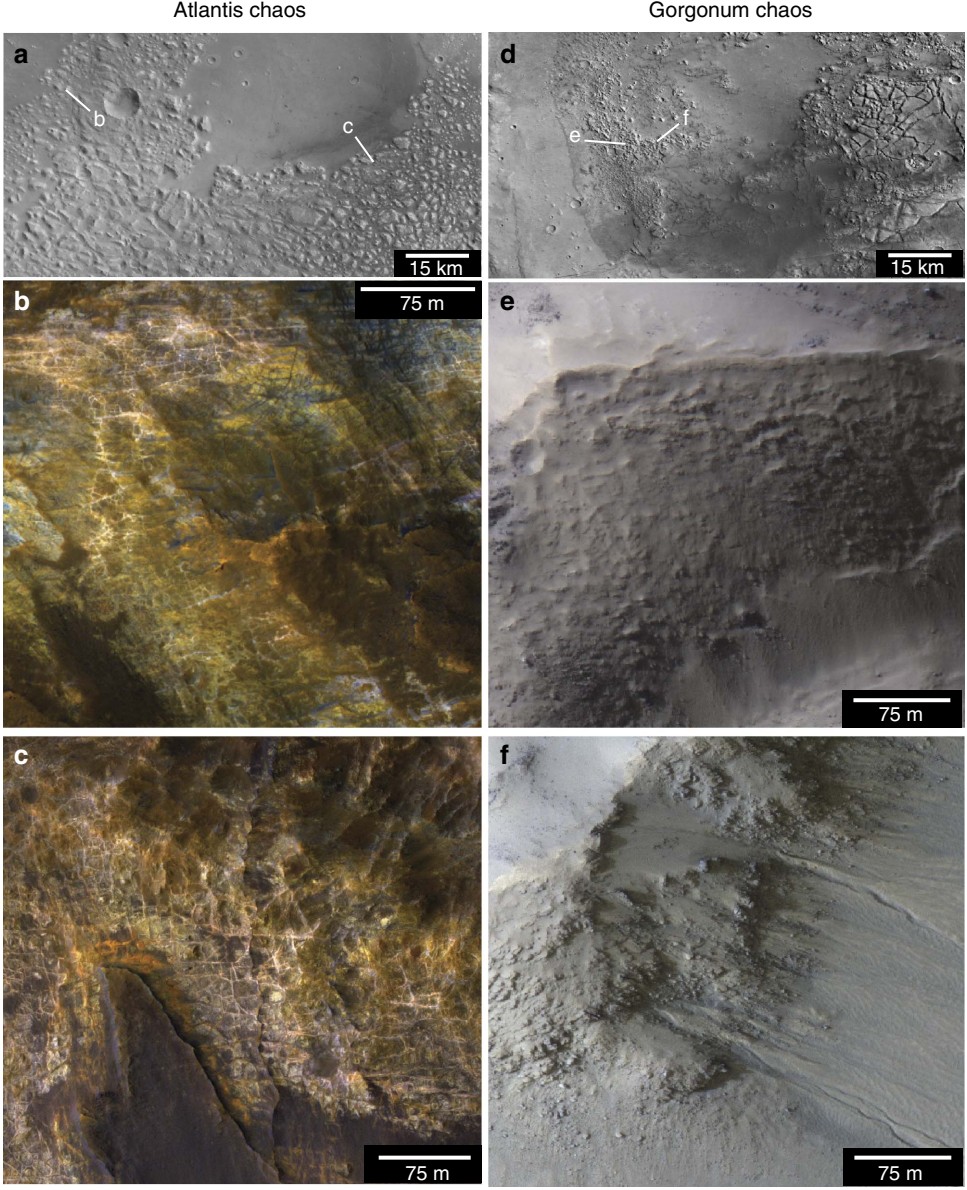

**Figure 3 | Sub-basins in Eridania contain multiple types of chaos-type terrain.** Deep basin deposits in the western unnamed basin, Ariadnes, Caralis and Atlantis (**a–c**) contain mottled bedrock with pervasive veins, strong spectral signatures of clay minerals and a complete lack of bedding. The east basins Gorgonum and Simois contain completely different units: younger, weakly layered, unaltered ash and lava deposits that have been fractured into chaos blocks (**d–f**). Context images in **a** and **d** are high resolution stereo camera (HRSC) panchromatic data. HiRISE data in **b,c** and **e,f** are IRB (infrared, red and blue-green). HiRISE, High Resolution Imaging Science Experiment.

~0.1–10 km diameter (Fig. 3). While these deep basin units are in some cases formally named 'chaos' and in other cases, 'colles[16],' there are some clear and important geological differences among the deposits that are not reflected in the naming convention and often confused in previous work (Fig. 3). Most importantly, the fractured blocks in the western and central parts of Eridania, as we argue in this paper, represent ancient, deep basin subaqueous units and those in the eastern parts of the basin are younger, eroded volcanics deposited subaerially.

Ariadnes Colles and Atlantis Chaos contain the best examples of deep basin deposits (Fig. 3a–c) that formed in deep water[16]. There, massive blocks of bedrock (lacking observable bedding)

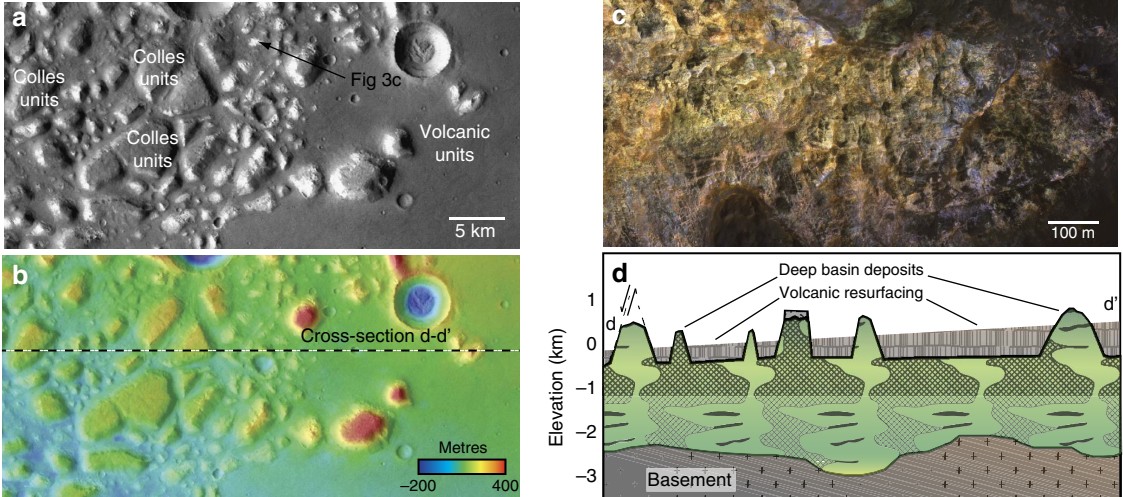

**Figure 4 | Geology of the deep basin deposits.** HRSC image (**a**) and HRSC elevation data (**b**) of Ariadnes colles with fractured, dismembered blocks of deep basin deposits that have been embayed and largely buried by younger volcanic deposits. HiRISE IRB data (**c**) show the unconformable relationship between the lower, ancient, pervasively altered and multi-coloured deep basin unit and the younger, relatively unaltered volcanics (**d**). HiRISE, High Resolution Imaging Science Experiment.

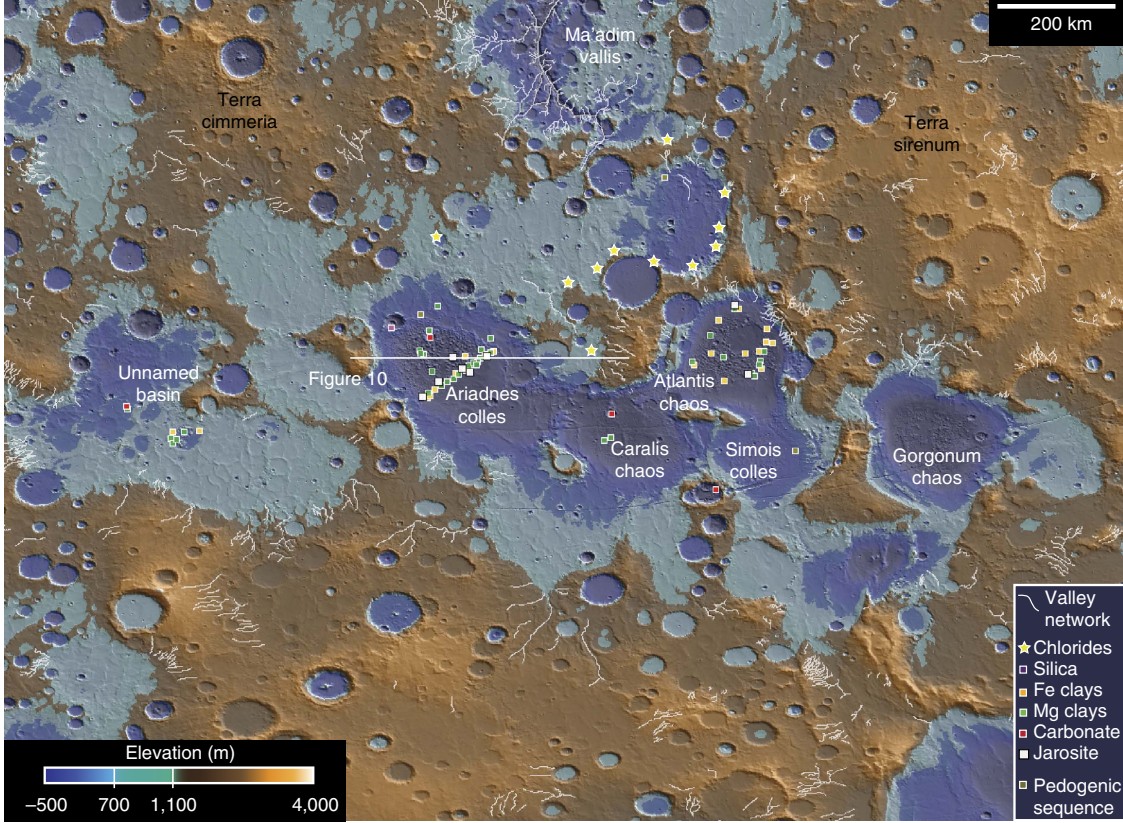

**Figure 5 | Topography and mineralogy of Eridania basin.** MOLA topographic data are colourized to show the maximum (1,100 m) and minimum (700 m) level of an ancient sea. Alteration minerals represent phases detected in this study using CRISM data with the exception of 'chlorides,' which were detected previously using THEMIS data. Deep basin units are pervasively altered to Fe- and Mg-rich clay minerals, and likely sulphides, which are traced by the occurrence of jarosite.

reach up to 400 m elevation above the surrounding dark-toned plains (Fig. 4). The deep basin unit was eroded and dismembered into buttes and mesas, subsequently embayed and blanketed by volcanic materials in the Hesperian, resulting in a kīpuka-like landscape (Fig. 4).

By contrast, block-forming units observed to the east in Gorgonum are completely different in terms of texture, bedding[17], colour and mineralogy (Fig. 3d–f ). These units are characterized by the presence of smaller blocks composed of a mixture of boulders and friable materials. Texturally, they are smooth and hummocky, and they have been widely eroded to form gullies in many cases[18], which is rare in the deep basin deposits to the west except where mantling volcanic deposits occur. The younger chaos units never show mottled colour patterns and do not contain evidence for fractures and veins. These are eroded blocks within the younger, superposed volcanic material (likely both ash and lava).

The block-forming basin unit in Gorgonum is substantially younger than the deep basin units in Ariadnes or Atlantis. Crater counting was performed within the deep basin deposits to estimate minimum ages for those deposits. We counted craters with diameters $\geq 500$ m throughout the deep basin deposits using Mars Context Imager (CTX) data as the base. The key result is that that deposits in Ariadnes and Atlantis basins are much older than basin deposits in eastern basins, especially Gorgonum. Assuming a crater producton function from Ivanov (2001) and absolute chronology based on Hartmann and Neukum[19,20], we estimate that minimum exposure age for the Ariadnes deposits at 3.77 Ga and the Gorgonum deposits at 3.47 Ga (Supplementary Fig. 2). These ages are consistent with previous results, which suggest that the Eridania basin-forming impacts occurred >4 Ga, the sea existed in the Late Noachian and was resurfaced by subaerial volcanism in the Late Hesperian[5,12,13,16]. A key new conclusion is that, while all of the Eridania sub-basins likely contained deep water environments, the deposits representing those environments are only well exposed in the western basins. They have been too intensely resurfaced in the east basins.

**Mineralogy of the Eridania basin.** In this work, we analysed the infrared spectra of all Compact Reconnaissance Imaging Spectrometer for Mars (CRISM) and all High Resolution Imaging Science Experiment data within the Eridania basin in order to evaluate the detailed mineralogy and geological context of deep basin deposits within the Eridania region (Fig. 5).

CRISM spectra acquired of the kīpukas throughout the western and central Eridania basin contain absorptions at ($\lambda$) $\sim 1.4$, 1.9

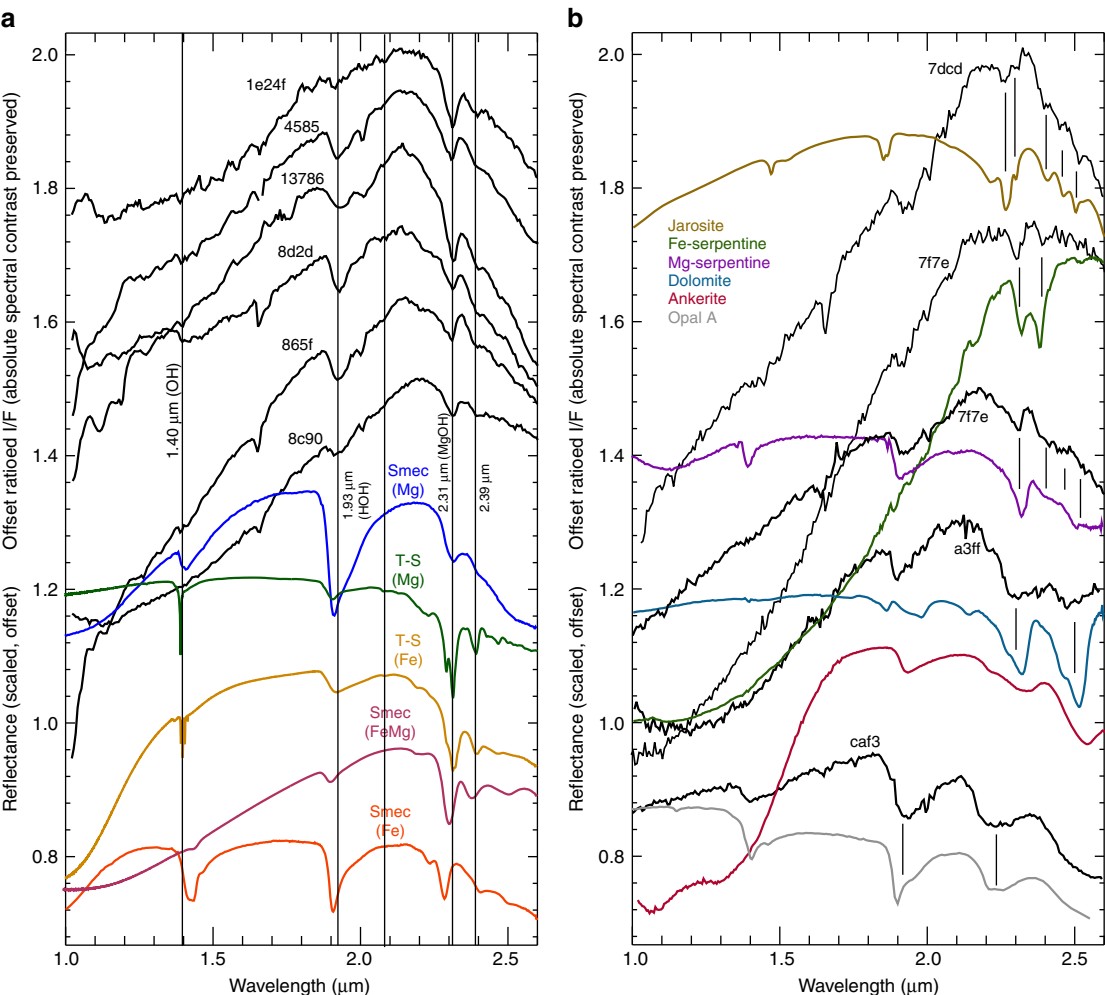

**Figure 6 | Mineralogy of the colles deposits.** CRISM I/F ratio data scaled and offset reveal a large suite of alteration minerals in the Eridania deep basin deposits. Most of the surfaces contain strong MgOH absorptions located near 2.315 μm and OH overtones located at 1.39–1.41 μm (**a**). Many also contain HOH absorptions located near 1.91 μm related to adsorbed water. Some exposures show evidence for a strong 1–2 μm slope suggestive of abundant Fe$^{2+}$ and various vibrational overtones related to MgOH, FeOH, MgFeOH, SO, SiOH and CO in serpentine-group minerals, jarosite, carbonates and silica (**b**).

and 2.3 μm indicative of the presence of Mg-rich and Fe-rich clay minerals[21,22] (Fig. 6). Fe-Mg-rich clays are common on Mars[23,24], but in detail, the deep basin bedrock units show spectral characteristics unusual for the planet.

CRISM spectra of the kīpuka blocks typically show absorptions at 2.31–2.315 μm characteristic of Mg-rich, trioctahedral clay minerals. Specifically, this absorption is indicative of $Mg_3OH$ and $Mg_2FeOH$ combination bands in the octahedral sheets of saponite, talc, serpentine and various mixed-layer clays[25]. Pure talc (that is, with little $Fe^{2+}$ or $Fe^{3+}$ substitution for $Mg^{2+}$) exhibits pronounced doublet absorptions at 2.29 and 2.31 μm, as do well-ordered examples of saponite and sepiolite.

The deep basin bedrock units show, in some cases, evidence for the doublet at 2.31–2.315 μm attributable to talc or other well-ordered, Mg-rich tetrahedral-octahedral-tetrahedral (TOT) clays such as sepiolite and some saponite, and in others, simply show a sharp absorption that could be attributable to saponite or Fe-rich talc. However, saponite and talc are commonly interstratified at the lattice scale in some seafloor settings[26,27] (mixed-layering), and some of the best matches to these Martian spectra correspond to spectra of mixed-layer seafloor clays on Earth[28]. In addition, the presence of a 1.9 μm $H_2O$ absorption in many of the detections (Fig. 6) suggests the presence of TOT clays with expandable layers (smectite or smectitic mixed-layer clays). However its absence in other materials suggests non-expandable TOT clays, such as talc, or that expandable clays have been locally dehydrated while others remain hydrated. Absorptions at 1.39, 2.315, 2.43 and 2.51 μm in some deposits suggest the presence of serpentine[29], which could also include serpentine-smectite mixed-layered clays (Fig. 6).

Fe-rich phyllosilicates are also observed. These deposits show absorptions at 2.295–2.305 μm, which are characteristic of Fe-rich dioctahedral mica or smectite with some Mg-substitution (that is, $^{VI}Fe^{3+}/Mg^{2+}$ molar ratio $\leq 4$)[28]. Such materials are spectrally similar to Fe-rich seafloor deposits sampled on Earth, and easily distinguishable from Al-bearing nontronite formed in a subaerial/continental setting[30]. In some deposits, an unusual doublet absorption at 2.32 and 2.38 is observed, and the same spectra display a weak or absent 1.9 μm feature and a very strong spectral slope from 1 to 2 μm, indicative of Fe-rich serpentine- or chlorite-group minerals[31]. In fact, the 1–2 μm slope, which is stronger than is typically observed in Martian clays[23] is likely a reflection of the abundant $Fe^{2+}$ present in many of the clay detections[32], and is a key indicator that the formation conditions likely involved a very Fe-rich fluid.

While the signature of phyllosilicates dominates the spectral character of the deep basin colles and chaos units, there are also several detections of jarosite occurring along with clay minerals within the chaos blocks (Fig. 6). The key distinguishing features of jarosite are absorptions at 2.265, 2.41, 2.46 and 2.51 μm (ref. 33). HOH absorptions occurring from 1.85 to 1.9 are variable on Mars[34] likely due to $K^+$, $Na^+$ and $H_3O^+$ content[33]. The most common formation mechanism for jarosite is through oxidative chemical weathering of sulphide minerals[35]. Sulphides might be present in the deep basin units, but they are very difficult to detect directly because, in the near infrared, they exhibit few or no distinguishing features. In fact, jarosite formed through oxidative weathering[36] is commonly considered a proxy for sulphide-bearing ore deposits[35].

Spectra extracted from the central peaks and interior walls of impact craters occurring within the basin centres provide information about the mineralogy of the units stratigraphically below the Mg-clay units (for example, Fig. 4d), and other deposits at depth that are poorly exposed. A 17 km diameter impact crater in Caralis Chaos contains structurally and texturally complex blocks of exhumed bedrock in its central uplift that display

spectral absorptions at ($\lambda$) ~2.31 and 2.51 μm indicative of Mg-rich or Fe-Mn-Ca-Mg carbonate[37] (Fig. 7). These materials also contain dense networks of crosscutting veins at the same scale (5–20 m-spacing) as is observed in the Mg-clay-bearing colles units (Fig. 7). Similarly, the central peak of an unnamed 15 km diameter crater in Ariadnes colles contains spectral evidence for Ca/Fe-carbonate in its central peak, as does a crater in the western, unnamed basin and the walls of a crater near Simois Colles where carbonates might have been mobilized within younger gullies[38].

The carbonates might have formed from impact-generated hydrothermal activity, but they appear to occur within coherent bedrock exposed in the uplifted peak and rim (Fig. 7). Depth of exhumation can be assessed assuming that the central peak uplift represents a depth 10% of the final crater diameter[39] ($D = 15$–17 km in this case). The carbonates were likely uplifted from 1 to 2 km below the floor of the basin where the impact occurred, which has a Mars Orbiter Laser Altimeter (MOLA) elevation of −50 m. This observation suggests that alteration is present to substantial depth within the basin, the deep basin units might be kilometres thick, and that the phyllosilicate-rich deep basin units might overly or be interbedded with carbonate-rich rocks.

One ~10 km diameter impact crater in Ariadnes Colles has exhumed material in its ejecta and rim that contain relatively strong absorptions at 1.9 and 2.24 μm indicating the presence of hydrated silica. But this crater is nested within a larger impact structure and it is possible that it has exhumed hydrothermal crater floor deposits formed from the earlier impact (Fig. 5).

Previous researchers identified chlorides in the region[40] using data from the Thermal Emission Imaging System (THEMIS) (Fig. 5). In contrast to the Mg- and Fe-rich clays, which are concentrated below 300 m MOLA elevation, the chlorides occur at higher elevations (350–1,050 m) (Fig. 8). The average elevation of chlorides is 660 m, which is similar to the low-water level of the Eridania sea (700 m)[13], suggesting they may have formed through evaporation in shallow seawater near the basin margin. Impact

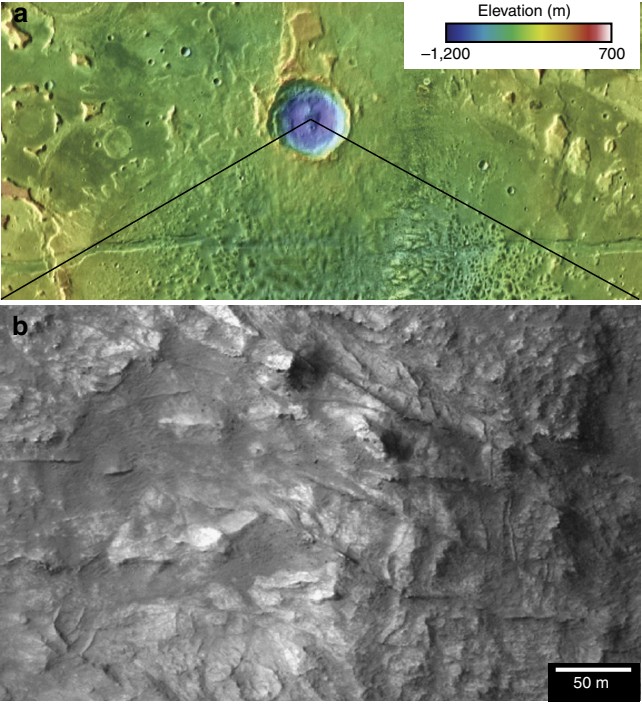

**Figure 7 | Context of carbonates in Eridania basin.** A 17 km diameter impact crater in Caralis chaos (**a**) contains carbonates within exhumed bedrock in its central peak (**b**) exhumed from 1 to 2 km depth.

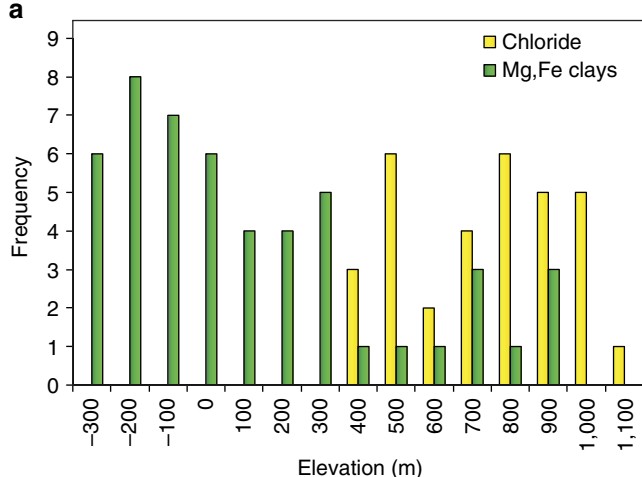

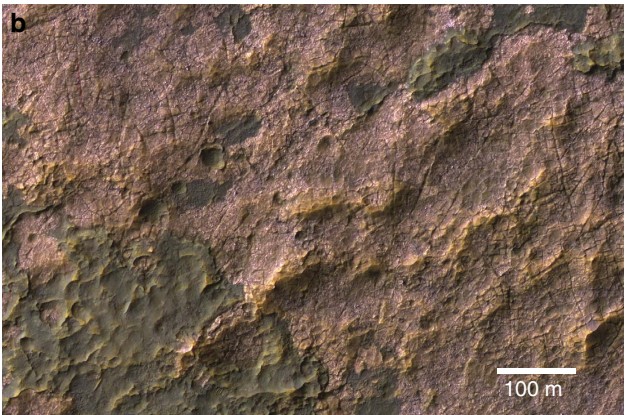

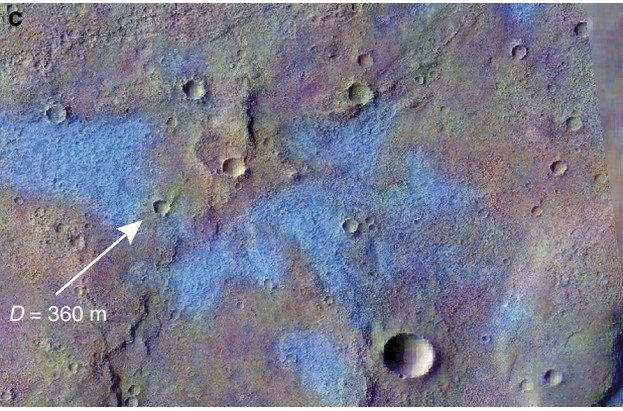

**Figure 8 | Characteristics of chloride deposits in Eridania basin.** Analysis of the MOLA elevation of coherent clay mineral exposures ($N = 50$) and chlorides ($N = 32$) within the basin shows that chlorides are only found at high elevations (350–1,100 m) while the clay deposits are concentrated at lower elevations (**a**). HiRISE IRB data showing the chlorides as pink in false colour reveal their fractured surface from beneath overlying dark volcanic unit (**b**). A THEMIS decorrelation stretch image (bands 8-7-5) showing chlorides in bright blue, overlaid on CTX data, shows that a 360 m-diameter crater has punctured through the chloride unit (**c**). HiRISE, High Resolution Imaging Science Experiment. MOLA, Mars Orbiter Laser Altimeter.

craters as small as 360 m-diametre have chloride-poor ejecta at distances up to 2–3 crater radii, suggesting that the craters have penetrated through a relatively thin (likely <30 m thick) chloride deposit (Fig. 8c).

Some of the eastern and northern parts of the basin contain Fe/Al-rich clays that are interpreted as pedogenic weathering sequences[21], as has been observed elsewhere on Mars[41,42]. These deposits typically contain 2.28–2.29 μm FeFeOH and FeAlOH absorptions corresponding to aluminous nontronite, and in two cases contain absorptions at 2.2 μm corresponding to kaolinite-group clays and a broad absorption from 2.3 to 2.5 μm likely interpreted as polyhydrated sulphates[43]. In Eridania, these deposits occur in the superposed volcanic resurfacing unit and are therefore younger than and of different context to the deep basin deposits.

In previous works, many of the pedogenic-type deposits have not been delineated from the true deep basin deposits. Here we point out that the deep basin deposits are clearly distinguishable based on texture, colour, stratigraphic relations and mineralogy. The older deep basin deposits in the western basins contain clear and strong evidence for complex Mg- and Fe-rich clay mineralogy. The younger volcanic resurfacing units (Fig. 3d–f) are generally spectrally unremarkable. But it is in association with these units that pedogenic-type sequences are found.

**Origin of deep basin deposits.** The deep basin units in Ariadnes, Atlantis and Caralis basins formed in association with significant amounts of water, as evidenced by the presence of 100 s of metre thick deposits of phyllosilicates containing dense vein networks. It is possible that clay in Eridania could have formed in an alkaline-saline evaporative lake setting[16,43], but we present challenges to the evaporite hypothesis, the most significant of which is the fact that the deep basin deposits are dominated by silicates rather than salt deposits.

The chemistry of the Eridania sea is unknown, though the volume to watershed ratio argues strongly that the sea was fed by groundwater[13,15]. Such a fluid would have been a $Fe^{2+}$, $Mg^{2+}$, $Ca^{2+}$, $Cl^{-}$, $HCO_3^{-}$ and sulphur-rich-brine after interaction with the regional mafic-ultramafic crust[44]. Evaporation of such a fluid could initially produce Fe-carbonates. In fact, the occurrence of carbonates, exhumed from depth within the deep basin deposits is consistent with precipitation from an early phase of evaporation or freezing. But continuation along an evaporative pathway would quickly exhaust $Fe^{2+}$ in solution leading to Mg-sulphate and ultimately to chloride precipitation[45], which is not observed in the deep basin units. The deep basin deposits (that is, in the basin centres) contain >400 m thick clay deposits and no detectable hydrated sulphates or chlorides (Figs 5a and 8a), though anhydrite and small amounts of hydrous salts are possible.

Chlorides are present at higher elevations along the interior basin margins at concentrations 10–25% by volume[46] and likely trace evaporitic, shallow water (<100 m) settings[40]. However, these deposits do not include any hydrated sulphates that should have precipitated before the chlorides during the evaporation sequence[44,45]. In most terrestrial playas, chloride deposits are situated in the middle of the basin rather than on the edges.

The lack of Mg, Fe sulphates in these deep basin deposits makes an evaporite-playa origin untenable. However, the chloride deposits on the basin margins may be related to evaporation in coastal, shallow water environments[47]. If the majority of the Eridania sea did not evaporate or freeze (both produce similar evaporite-type deposits), then the fluid was likely lost back into the subsurface due to some fundamental change in the regional groundwater table, perhaps including the formation of a new, deep basin that affected groundwater flow.

A detrital origin of clay-rich, deep basin deposits in Eridania is also unlikely. The concave topography of the deep basins below 700 m elevation is unusual for Martian basins[13]

(see Supplementary Fig. 2), most of which have flat floors that formed through subaerial resurfacing. The shape of Eridania basin floors argues strongly that the surfaces were protected below water[13] during the period of intense sedimentary deposition on Mars (Late Noachian)[11,48].

The deep basin deposits are unlikely to have formed by air fall as has been previously concluded[12,16]. The younger Electris deposits, which are layered, are of consistent thickness throughout the region (150–200 m), and occur at a wide range of elevations are consistent with an air fall origin[49]. By contrast, the deep basin deposits in Ariadnes and Atlantis are not layered, are thick, and concentrated at low elevations (Figs 5 and 8a). Ariadnes Colles and Atlantis Chaos contain at least $1–5 \times 10^4 \, \text{km}^3$ of altered material (assuming a minimum thickness of 400 m and a likely thickness of >1 km). Deposition of such a thickness and volume of material is possible proximal to explosive vents[50]. It is possible that unrecognized volcanic vents are present[51], but any air fall origin fails to account for why such thick deposits are found within the basins, but no trace of similar deposits of similar age are found outside the basins. Even if air fall deposition cannot be ruled out as a geological process, this model seemingly requires major volcanic source regions near, but outside the basins while ignoring the fact that volcanism would most likely be localized in the basins themselves, as is observed elsewhere on Mars and on other planets[52].

The most plausible way to produce such large volumes of deep basin, deep water deposits is through seafloor volcanic-sedimentary processes focused in the basin floors where fractured, thinner crust and higher heat flow would be expected. Large volumes of Hesperian lava present throughout Eridania are proof that significant volcanism occurred within the basins. We argue that this volcanism did not suddenly begin after the sea had ceased to exist in the Early Hesperian, but most likely began in the Noachian, shortly after the basins formed. A sea of the size of Eridania is unlikely to have been ephemeral and therefore, it is nearly inescapable that subaqueous volcanism would have occurred during the period in which the sea existed.

Previous authors have demonstrated that most ancient, large impact basins on Mars were resurfaced by ultramafic to mafic volcanic materials—olivine rich lavas that have erupted through the relatively thin crust of basin environments[53]. The Eridania basins would have likely had the same type of activity. The important

difference in Eridania is that a deep sea was present while volcanism occurred. If the Eridania sea level was at the 700 m elevation level (a conservative estimate), it implies that the deep basin deposits formed beneath 500–1,200 m water depth (Fig. 9). The lower gravity of Mars results in lower water pressure in a Martian sea compared to one on Earth, for a given depth (Fig. 9). Seafloor volcanism in Eridania would have occurred at water pressures of 20–50 bars. At these pressures, Martian seafloor volcanism could have included both effusive and explosive elements, in addition to chemical sedimentation from hydrothermal fluids (Fig. 10). The transition from altered deep basin deposits to flood lavas in the Hesperian does not represent the onset of volcanism in the basins, but the transition from subaqueous to subaerial volcanic activity as the Eridania sea came to an end.

**Implications of the hydrothermal seafloor model**. We conclude that thick, massive, clay-, carbonate- and likely sulphide-bearing deposits in Eridania basin formed in a deep-water hydrothermal environment on ancient Mars (>3.8 billion years ago) (Fig. 10). Saponite, talc, talc-saponite, Mg-bearing nontronite, glauconite, serpentine and berthierine are all common in terrestrial seafloor deposits[26,27,54]. The clay assemblages and spectral trends observed in seafloor deposits on Earth provide a good analogue for the deep basin deposits detected remotely in Eridania[28]. Salts only observed at higher elevations likely represent coastal evaporative settings (Fig. 10). Several lines of evidence strongly suggest that Eridania was a sustained inland sea in the late Noachian.

The deep-water environment was likely reducing based on direct evidence for $Fe^{2+}$-rich clay minerals and indirect evidence for Fe-sulphides. This could be an indication of stratification of an ancient sea beneath an oxidized atmosphere, chemical isolation in an ice-covered sea, or quasi-equilibrium with a reduced atmosphere. The ancient Eridania sea deposits might represent a setting analogous to Fe-rich sea environments present on the early Earth.

Ancient, deep-water hydrothermal deposits in Eridania basin represent a new category of astrobiological target on Mars. To date, the search for habitable environments on Mars has been focused on exploration of ephemeral playa and shallow lacustrine settings. The Eridania deposits represent an ancient environment rich in chemical nutrients and energy sources. Such a deep-water

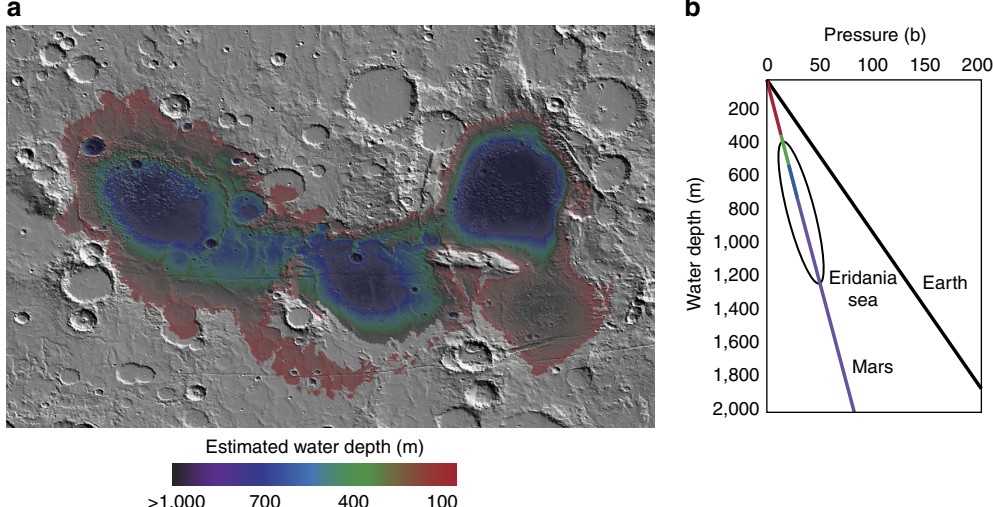

**Figure 9 | Relationship of water depth to seafloor pressure on Mars.** Using a conservative estimate of extent of the Eridania sea (700 m contour), the depth of the sea can be estimated. Most deep basin deposits likely formed in deep water (400–1,200 m) (**a**). Water pressure at this depth (∼20–50 bars) would be less than that on Earth due to the lower gravity on Mars (**b**). The pressure at this depth would have affected the style of volcanism.

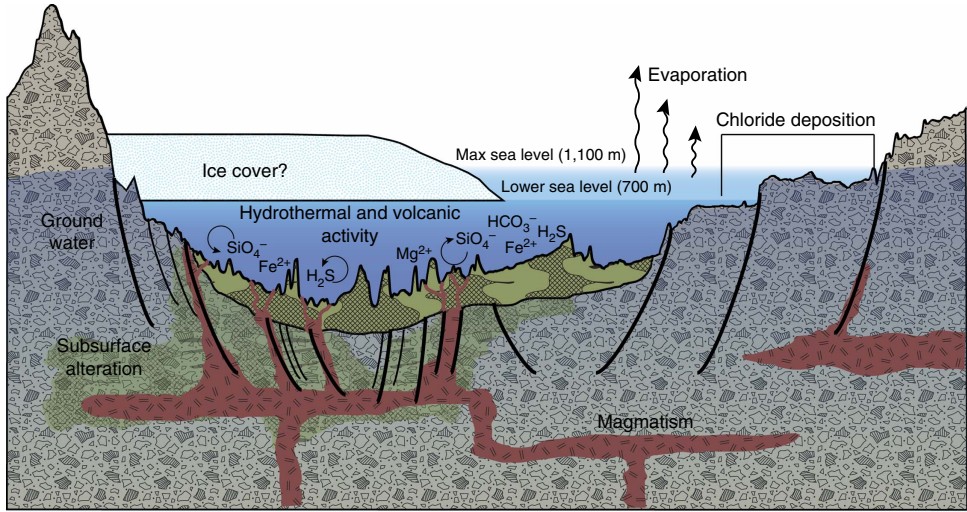

**Figure 10 | A geologic model for Eridania basin on ancient Mars.** This diagram shows an exaggerated MOLA topographic profile of a 450-km-long transect through Ariadnes basin (see Fig. 5 for context). Water levels are coloured to match the minimum (700 m) and maximum (1,100 m) elevations of the sea level in Fig. 5. Thick, clay-rich deposits formed through hydrothermal alteration of ultramafic volcanic materials in deep water. Deep-seated structural discontinuities could have facilitated the ascent of magma from a mantle source. Chloride deposits formed from evaporation of seawater at higher elevations in the basin.

environment would have been protected from harsh surface conditions and ideally suited for preservation of organic matter under reducing conditions. In fact, the earliest evidence of life on Earth seemingly corresponds to seafloor deposits[2] of similar origin and age, although the terrestrial counterparts are metamorphosed and metasomatized. Eridania seafloor deposits are not only of interest for Mars exploration, they represent a window into early Earth.

**Data availability.** Data required to complete this work include: (1) hyperspectral image cubes from CRISM; (2) high-resolution visible images from High Resolution Imaging Science Experiment; (3) day and night-time infrared image data from THEMIS; (4) visible image and digital elevation information from HRSC; and (5) topographic data from MOLA. All data used in this work, as well as the software used to process CRISM data, are available through the Planetary Data System (https://pds.nasa.gov). Other planetary data sets and visualization capabilities are available within the JMARS software provided by Arizona State University (https://jmars.asu.edu).

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

## Author contributions

J.M. carried out all spectral and geological analyses, and wrote the paper. E.Z.N.D. contributed the initial vision for the project. P.B.N. contributed to early phases of the project. J.C. helped write the manuscript and contributed expertise about seafloor processes.

## Additional information

**Competing interests:** The authors declare no competing financial interests.

