## [Peer Review File · Nature Communications]

Reviewers' Comments:

Reviewer #1 (Remarks to the Author)

The manuscript by Michalski et al. proposes that the lakes of the Eridiani basin, Mars, experienced enhanced hydrothermal activity due to interactions between water and volcanic activity during the Noachian period. This activity is compared to terrestrial hydrothermal activity. The manuscript proposes that Mars' alteration at Eridiani is a good analogue for the Archean Earth hydrothermal alteration.

Overall, I find the studied region fascinating and it deserves a strong interest. The synthesis provided in this paper could be of great help for the community, but the authors should better present what is part of the synthesis, and what is original to their work. The scenario proposed is a fair explanation for the mineralogical assemblages proposed, but it should be demonstrated more in depth to be convincing, especially extending the two short paragraphs where it is proposed, discussing why effusive lava flows would be possible below shallow water on Mars and identifying evidence for this early volcanism. The outcome as an analogue for Archean seafloor is a nice idea, but by far excessive given the differences in the style of volcanism and the Fe-S-richer content of Mars bulk composition. So, I am not against the eventual publication of this paper, but it should be strongly modified before.

Detailed comments:

The manuscript starts with descriptions of Eridiani basin, namely context, geomorphology and mineralogy. In the introduction, I do not understand the statement: "consistent with regional magmatism and the presence of a high-potassium anomaly could be an indication of a deep mantle source for ancient volcanism in the area ». Why high potassium anomaly could be an indication of deep mantle, it could also be related to ancient pieces of crust, as observed at Gale crater by Curiosity. Speculations like these in the introduction is difficult to follow and should better be in the discussion.

In the three following sections, the paper should better separate the new results provided by the manuscript from the review of observations already acquired on the Eridania basins in previous studies. For instance, the section "Evidence for an ancient sea in Eridania basin" is a mini-synthesis of previous work, not an original result. Actually, it forgets the first of these articles : Irwin et al. 2002 at Science; who first discovered these series of lake. In this section of the synthesis, the text also refers to Bouley et al. (2016) as: « ...more importantly into the centre of a band of mild climatic conditions in the southern highlands ». Given that Bouley' paper proposes glaciers/snow having formed valley networks in the southern equatorial belt, this may not correspond to a mild climate at this location at all. It could have been warmer elsewhere. To be clear, the article by Bouley only considers precipitations (as snow), not the climate conditions. So the review of past studies in this section should also be revised.

The section entitled: "Multiple types of colles and chaos units in Eridania basin" also refers to many papers. It is a nice review, and refers adequately to these papers, but it does not explain what original results are proposed in the article. In this section, Figure 2 is not enough described. What about the boxwork/veins of figures 2b and 2c? Boxworks could form by diagenesis within an aquifer and not only below lake or sea (see Metz and Grotzinger, 2013 for more on boxwork textures).

The section « Mineralogy of the Eridania basin » is the most detailed observation section. The description is well done and mineral assemblages are very interesting, but, here again, most minerals have been detected earlier to this article. It is really difficult to identify which correspond

to new detections and are original results from what has been published previously. For instance, a table with previous detections and corresponding locations/CRISM cube would help, and/or identification of those cubes on the map of Figure 1.

After this review in three steps, the section "Origin of basins", line 229 to 305 is the original part of the manuscript. Within 70 lines and one figure this section tries to propose a scenario for the formation of the mineral assemblages at the bottom of the basins as due to hydrothermal interaction with volcanism. It is possible that volcanism may have triggered this variety of assemblages, but the demonstration itself is rather short and unprecise. The actual scenario proposed extends over 15 lines from line 291 to 305; the text before discards other hypotheses. I would have better started with that part of the section, to demonstrate the scenario more extensively, and then discards other hypotheses. As it is, this section should be the heart of the paper but it is not enough convincing. For instance, where are the presumed Noachian lava flows? Are they still visible in the morphology? Lava flows would have formed below water, so why is there no explosive volcanism visible? On Earth, the thickness of water above seafloor provides the possibility to have subaqueous effusive lava flows. It is by far not obvious that subaqueous effusive lava flows are possible on Mars with the relatively thin (several 100s of meters) depth of water at Eridiani basins.

Explosive volcanism is actually discussed before the proposed scenario, but the corresponding 15 lines are spent to propose the presence of airfall deposition from other regions, and then discard it. However, I would not wait for airfall deposits to be so variable in mineralogy; I would rather question the role of lava-water interaction at Eridiani itself.

In the Supp material, some additional evidence provided to reinforce the interpretations are also matter of discussion. It is written line 76 of Supp Mat.: "The concave structure of the Eridania basins is an indication that, during the only intense period of erosive activity in mars history, these basins were protected beneath water". I concur with this idea, but then why volcanism would have taken place ? I mean the other basins are filled by volcanic episodes and do not display the same topography. These other basins are filled by Hesperian lava flows after any episode of lake. So the comparison could be interpreted as a proof that Hesperian volcanism, although present at Eridiani, did not cover all basins contrary to those other basins which are now flat. This does not prove that Eridiani basins are not filled by sediments only.

In the section comparing Eridiani basin with Earth, I am uncomfortable with the term « sea », as used for « hydrothermal seafloor », given that it is specifically used on Earth for oceanic crust that experience hydrothermal alteration linked to the high geothermal flux around mid-oceanic ridges due to plate tectonics. The context proposed for Eridiani basin is mostly lacustrine, with some interesting alteration going on its floor. The implications for early Earth are also exaggerated, given the different chemistry of Mars (more Fe and S).

Reviewer #2 (Remarks to the Author)

My comments to the Authors are in the attached PDF file.

Reviewer #3 (Remarks to the Author)

General comments:

This paper describes the geology and the mineralogy of Eridania Basin using the data acquired by CRISM, HiRISE and CTX instruments. Purpose of the manuscript is to demonstrate that deposits in Eridania formed in a hydrothermal seafloor setting. Although I'm not a geologist, I believe that the paragraph about the geology is described adequately and I appreciated the general discussion done. Regard to the mineralogical discussion, most of the information needed to understand the presence of various mineral are given, but in some cases (e.g. jarosite) the presence of the mineral is not yet sufficiently proved. For this reason the author should at least address the points listed below. Overall, the text is clear and accessible to nonspecialist. If these points will be clarify

my recommendation is for acceptance.

Specific comments are found below. Review by F.G. Carrozzo

Line 58. In figure 1, it is not clear where is Eridania Basin. In the figure are present various regions such as Gorgonum Chaos and I do not understand if these areas are located in Eridania Basin. The author might consider to draw the boundaries of Eridania using a curve.

Lines 106. The author cites supplementary materials referring to the Figure S-2. In the caption remove "for are shown" and add "are shown" after "Gournum (d)".

Line 113. Where the kipukas are located? I suggest to add a reference to a figure or to add a new figure. The term is specialistic and a figure can help the reader to understand its nature.

Line 128. Talc has a set of absorptions (1.44, 1.53, 2.0-2.18) not present in the CRISM spectra of figure 4. It is useful to show an example of talc spectrum. Why the author cited the talc? I believe that the yellow spectrum of Figure 4a is saponite. Can the author explain where the T-S library spectra have been taken? What are the samples?

Line 143. The author cited mica as possible candidate due to the absorption at 2.295-2.305 μm . Micas showing this band (e.g. glauconite) have other absorptions at $\sim 1.91 \mu\text{m}$ and at 2.36-2.37 μm that are not present in the spectra of figure 1a. How does the author explain it?

Line 147. Other minerals have the same doublet (e.g. mica or amphibole). Why these minerals are excluded?

Line 159. The author cited diagnostic features useful to detect jarosite. From the figure 4b it is difficult to distinguish the position of the minima. May the author show me a comparison of the main bands using a close up (between 1.8 and 2.5 μm)? In my opinion the band at 2.46 is not present in the CRISM spectrum.

In the same figure it is present a line near 2.3 μm showing the presence of the spectral feature in both CRISM and library spectra, but why the band at 2.21 μm is not discussed? In my opinion, sentence at lines 159-161 does not justify the absorption because it is at longer wavelengths. In the CRISM spectrum the band at 1.85-1.9 μm , diagnostic of jarosite, is not present. It is after 1.9 μm . Can the author explain better why there is this difference?

Line 170. Figure 4D is not present.

Paragraph Carbonates, Line 167. The CRISM spectrum a3 (in figure 4b) has a broad absorption near 1.9 μm that is not present in the library spectrum. This band and the other minima can be consistent with the presence of hydrated phases, for example a mixture of carbonates (e.g. calcite, magnesite) with Mg-phyllsilicates (e.g., vermiculite, saponite). The author should discuss it as suggested for example in Carrozzo et al. 2017 (Icarus, "Geology and mineralogy of the Auki Crater, Tyrrhena Terra, Mars a possible post impact-induced hydrothermal system").

Line 174. Why the author show the spectrum of ankerite? The minima of absorptions of spectrum a3 are different.

Paragraph Chlorides, Lines 198-208. The discussion about the absorptions of chlorides is not present. Moreover, in figure 4 the spectra are not present.

Lines 222-225. I'm not sure that the mineral spectra cited are present in figure 4. If not present, the comparison of these CRISM spectra with library ones is useful.

Reviewer #1 (Remarks to the Author):

The manuscript by Michalski et al. proposes that the lakes of the Eridiani basin, Mars, experienced enhanced hydrothermal activity due to interactions between water and volcanic activity during the Noachian period. This activity is compared to terrestrial hydrothermal activity. The manuscript proposes that Mars' alteration at Eridiani is a good analogue for the Archean Earth hydrothermal alteration.

Overall, I find the studied region fascinating and it deserves a strong interest. The synthesis provided in this paper could be of great help for the community, but the authors should better present what is part of the synthesis, and what is original to their work.

We have modified the text to more clearly state what is review and what is original. Modifications to the text are shown in green font.

The scenario proposed is a fair explanation for the mineralogical assemblages proposed, but it should be demonstrated more in depth to be convincing, especially extending the two short paragraphs where it is proposed, discussing why effusive lava flows would be possible below shallow water on Mars and identifying evidence for this early volcanism. The outcome as an analogue for Archean seafloor is a nice idea, but by far excessive given the differences in the style of volcanism and the Fe-S-richer content of Mars bulk composition. So, I am not against the eventual publication of this paper, but it should be strongly modified before.

We appreciate the reviewer's comments and have added text to explain why this interpretation is not "excessive." First, to address the question regarding the differences in style of volcanism between the Earth and Mars, we wish to point out that most of Mars is covered by (eroded and modified) basaltic volcanic materials that likely erupted from fissures and point sources (See Greeley and Spudis, 1982). While the bulk composition of Mars is thought to be much more Fe-rich than Earth, the crust of Mars seems to be largely basaltic (see a number papers by McSween et al., Taylor et al., and others). The styles of volcanism between the Earth and Mars contain significant crossover in the sense that effusive emplacement of lava has been important in both cases.

However, there are some key differences in terms of volcanic processes between Earth and Mars, mostly related to differences of pressure in the crust due to lower gravity on Mars. In general, basaltic volcanoes should be much more explosive on Mars than on Earth (see Michalski and Bleacher, 2013). But to address the question of how lava could be emplaced beneath deep water (~1000 m) in Eridania basin, we must consider the effects of gravity on water pressure as well.

It is certainly possible that the deep basin deposits in Eridania represent intensely altered lavas, but this is not necessarily essential to the main point of the paper. We mention in the paper that, given that many or most of large, ancient impact basins in the highlands have been resurfaced by olivine-rich lava (see Edwards et al., Icarus, 2014), it would be extremely surprising if these relatively deep basins in Eridania were not also resurfaced by volcanism. But – the key point here is that, when this activity seems to have occurred on Mars (3.6-4 Ga), Eridania actually contained a vast, deep sea. So, if volcanism also occurred here as it did elsewhere, then it would have been subaqueous.

Note that on Earth, the seafloor volcanism can include extensive lobate basaltic lava flows at a depth of >1 km. On Mars the same could have occurred. But on Mars, because of the lower gravity, the pressure exerted by a 1-km-thick column of seawater would actually be significantly lower. So, pressure at 1 km depth in a Martian sea would be ~40 bars compared to ~100 bars on Earth. We have added a new figure to illustrate this difference.

We suggest in this paper that the deep basin deposits are deep-water volcanics or hydrothermally altered materials (including hydrothermally altered impact glass sheets in the floors of the basins). Previous papers have hypothesized about the origin of deep basin deposits, but generally concluded that they are likely detrital, air fall materials or evaporates. We show in the paper that they are unlikely to be evaporites. And, we argue that the detrital model is inconsistent with the large volumes of altered floor deposits present (>1 x 10⁴ km³ per sub-basin), given the lack of similar materials in the highlands near, but outside the basins, and given the limited erosional potential of the valleys that contribute to the basin.

We wanted to take a moment to explain our perspectives here for clarity. We thank the reviewer for raising these issues. *Most importantly, we have modified the text to make these points more clearly in the manuscript itself.*

Detailed comments:

The manuscript starts with descriptions of Eridani basin, namely context, geomorphology and mineralogy. In the introduction, I do not understand the statement: “consistent with regional magmatism and the presence of a high-potassium anomaly could be an indication of a deep mantle source for ancient volcanism in the area ».

Why high potassium anomaly could be an indication of deep mantle, it could also be related to ancient pieces of crust, as observed at Gale crater by Curiosity. Speculations like these in the introduction is difficult to follow and should better be in the discussion.

We point out in the introduction that this region has a regional scale potassium anomaly. This is not speculation, but an observation. We have clarified the text to state that K-enrichment at this scale can be the result of volcanic resurfacing from deep mantle sources, or from widespread alteration of the crust. We added two references to help support the arguments and help the readers find more information on this topic if they seek to do so.

In the three following sections, the paper should better separate the new results provided by the manuscript from the review of observations already acquired on the Eridania basins in previous studies. For instance, the section “Evidence for an ancient sea in Eridania basin” is a mini-synthesis of previous work, not an original result. Actually, it forgets the first of these articles : Irwin et al. 2002 at Science; who first discovered these series of lake.

In the revised version, we have taken care to identify as clearly as possible what is synthesis and what is new. This is an important point and we certainly want to avoid confusion. The section on evidence for a sea in Eridania has been updated to clearly explain that previous researchers have made this argument for a sea based on geomorphic data. In the mineralogy section, we cite previous work, but we point out that in this work, we analysed ~150 CRISM cubes (all of the CRISM data in the basin), and all of the corresponding HiRISE data.

In this section of the synthesis, the text also refers to Bouley et al. (2016) as: « ...more importantly into the centre of a band of mild climatic conditions in the southern highlands ». Given that Bouley’ paper proposes glaciers/snow having formed valley networks in the southern equatorial belt, this may not correspond to a mild climate at this location at all. It could have been warmer elsewhere. To be clear, the article by Bouley only considers precipitations (as snow), not the climate conditions. So the review of past studies in this section should also be revised.

We have deleted this text.

The section entitled: “Multiple types of colles and chaos units in Eridania basin” also refers to many papers. It is a nice review, and refers adequately to these papers, but it does not explain what original results are proposed in the article. In this section, Figure 2 is not enough described. What about the boxwork/veins of figures 2b and 2c? Boxworks could form by diagenesis within an aquifer and not only below lake or sea (see Metz and Grotzinger, 2013 for more on boxwork textures).

Previous researchers have lumped all chaos or colles units together as “knobby”

terrain, or otherwise confused the terminology. That is one of the reasons that this paper is important, and the main reason we require this section of the paper. We have adequately cited previous work. All of the interpretations presented here are original. Yes, it is true that others have described the buttes in the basins, but the previous descriptions are not adequate to identify the key differences between chaos units of different origins and ages in the different basins.

We do not cite the boxwork veins as “smoking gun” evidence for the past existence of a sea, however, we do point out that these features only form when there is significant water in the system. That is also the conclusion in the Seibach and Grotzinger paper, though admitted in a diagenetic context. The Eridania features formed in the ancient seafloor and the elements of fluid flow that are “diagenetic” apply here.

The section « Mineralogy of the Eridania basin » is the most detailed observation section. The description is well done and mineral assemblages are very interesting, but, here again, most minerals have been detected earlier to this article. It is really difficult to identify which correspond to new detections and are original results from what has been published previously. For instance, a table with previous detections and corresponding locations/CRISM cube would help, and/or identification of those cubes on the map of Figure 1.

We appreciate the reviewer’s points here and have revised the manuscript to point out more clearly what is new and what is not new. To be clear, there is a significant amount that is new here in terms of both mineral detection and interpretation of geologic context of those minerals. For example, several authors have published on the CRISM-derived mineralogy of the region, but the most detailed information is contained in the dissertation and an Icarus paper, both by Lorenz Wendt. This is a great piece of work and we find the results insightful. However, in that paper, and in others, there are two key issues: 1) the light-toned, older deep-basin material is generally treated similarly to the light-toned, younger material (not deep-basin deposits) in previous work and 2) the mineralogy of the deep basin deposits (referred to as knobby terrain in most works) is generally described as Fe/Mg-clays). These differences are important for a few key reasons.

In this paper, we emphasize the difference in geologic context between the deep basin deposits and the other light-toned materials. The deep basin deposits have mineralogy dominated by Fe-rich materials and Mg-rich materials, but have no Al-bearing materials, no chlorides, and no pedogenic phases. The younger light-toned materials do contain these sorts of materials, such as kaolinite and gypsum.

With regard to the second point, we emphasize that the deep basin deposits are not only dominated by Fe-Mg clays, but that there is much more to the story that has not been discussed before. Previous works have not identified very Fe-rich phases such as berthierine and chamosite, have not identified serpentine, have not discussed the presence of talc and talc-saponite, and have not identified carbonates or amorphous

silica. All of those things are new in this paper

After this review in three steps, the section “Origin of basins”, line 229 to 305 is the original part of the manuscript. Within 70 lines and one figure this section tries to propose a scenario for the formation of the mineral assemblages at the bottom of the basins as due to hydrothermal interaction with volcanism. It is possible that volcanism may have triggered this variety of assemblages, but the demonstration itself is rather short and unprecise. The actual scenario proposed extends over 15 lines from line 291 to 305; the text before discards other hypotheses. I would have better started with that part of the section, to demonstrate the scenario more extensively, and then discards other hypotheses. As it is, this section should be the heart of the paper but it is not enough convincing. For instance, where are the presumed Noachian lava flows? Are they still visible in the morphology? Lava flows would have formed below water, so why is there no explosive volcanism visible?

These issues come back to the question of “what would submarine volcanism look like on Mars?” If the volcanism occurred beneath deep water, it would probably consist of flows and fragmented glass, along with chemical sediments deposited from hydrothermal fluids – as occurs on Earth. We would not expect to see the types of hydrovolcanic landforms that are observed in subglacial settings such as tuyas and mobergs.

The question of “where are the Noachian lava flows?” is a great one. These basins are deep - deeper than many other basins on Mars. At the time when they existed, other highlands and basins were being resurfaced by volcanism. It is hard to imagine that the floors of these basins would not have been resurfaced by volcanism at that time as well. But, the products of that ultramafic volcanism in this case is not olivine-rich lava, as is observed in many other cases; it is strongly Fe- and Mg-rich alteration. So, the question of where are the Noachian volcanics leads to one of several answers: a) they never formed (unlikely), b) they were buried by deep basin deposits, or c) the deep basin deposits are subaqueous hydrothermal-volcanic deposits.

On Earth, the thickness of water above seafloor provides the possibility to have subaqueous effusive lava flows. It is by far not obvious that subaqueous effusive lava flows are possible on Mars with the relatively thin (several 100s of meters) depth of water at Eridiani basins.

The question of whether volcanic flows were effusive within Eridania basin is not central to the point of this paper. The main point of the paper is that volcanism would have occurred in the basin in deep water. It is the hydrothermal mineralization associated with the volcanism that is most interesting. However, it is a fair question to wonder what the nature of that volcanism might have been like.

In order to address this question we have added a new figure (Figure 9) which shows estimated water depth in the basin and the relationship between water depth

and pressure for the Earth and Mars. Erwin and other have estimated the maximum height of the Eridania sea at 1100 m MOLA elevation, with a more stable level at 700 m elevation. Using the more conservative estimate and present day basin topography, we now show the likely water depth in the basins (see “a” below). Nearly all of the proposed volcanic deposits would have formed at a water depth of >400 m, and most of them would have formed at even greater depths (700-1000 m). Using an estimated density for the Eridania fluid of 1100 kg/m³ (similar to terrestrial seawater) and adjusting for Martian gravity, we then relate the pressure to water depth on Mars (see “b” below). As you can see, the Eridania volcanism would potentially have occurred at pressures of 20-50 bars. At these pressures, we would not expect to see phreatomagmatic activity, but perhaps volcanism more closely akin to terrestrial seafloor volcanism.

a

b

Explosive volcanism is actually discussed before the proposed scenario, but the corresponding 15 lines are spent to propose the presence of airfall deposition from other regions, and then discard it. However, I would not wait for airfall deposits to be so variable in mineralogy; I would rather question the role of lava-water interaction at Eridania itself.

Previous authors have favoured the air fall scenario, which is why we discuss this explicitly. As for the hydrovolcanic scenario questioned in this review, please see response to the previous question.

In the Supp material, some additional evidence provided to reinforce the interpretations are also matter of discussion. It is written line 76 of Supp Mat.: “The

concave structure of the Eridania basins is an indication that, during the only intense period of erosive activity in Mars history, these basins were protected beneath water“. I concur with this idea, but then why volcanism would have taken place? I mean the other basins are filled by volcanic episodes and do not display the same topography. These other basins are filled by Hesperian lava flows after any episode of lake. So the comparison could be interpreted as a proof that Hesperian volcanism, although present at Eridania, did not cover all basins contrary to those other basins which are now flat. This does not prove that Eridania basins are not filled by sediments only.

What we mean to say (and what Irwin et al. point out) is that the topography of most basins was dominated by volcanic resurfacing and subaerial erosion. In this case, the basin topography was preserved because during the period of intense subaerial erosion and valley network formation, these basins were protected by water (and potentially ice-covered water). We have hopefully clarified this issue in the text.

In the section comparing Eridania basin with Earth, I am uncomfortable with the term « sea », as used for « hydrothermal seafloor », given that it is specifically used on Earth for oceanic crust that experience hydrothermal alteration linked to the high geothermal flux around mid-oceanic ridges due to plate tectonics. The context proposed for Eridania basin is mostly lacustrine, with some interesting alteration going on its floor. The implications for early Earth are also exaggerated, given the different chemistry of Mars (more Fe and S).

Of course there is no “official” distinction between lake and sea, but please consider these points.

1 – The area of the Eridania water body is estimated to have been $1-3 \times 10^6$ km². A conservative estimate of the water volume is presented in Fasset and Head, 2008. In that paper, they describe the characteristics of all closed basin lakes on Mars (208 of them, including Eridania). They estimate the volume of water that would have been present in each basin based on the depth, shape and mapped shorelines or base level of each lake. They conclude that Eridania would have been the largest basin on Mars, and if you consider their published water volumes, they show that the volume of the Eridania water body is actually greater than the volume of all of the other lakes on Mars combined. We have created a new figure for this paper (Figure 2) that shows the scaled relationships in terms of spherical volume of a conservative estimate of the Eridania sea compared to all other martian lakes, as well as Earth’s Mediterranean sea, Caspian Sea, Great Lakes, and Lake Baikal. As you can see, the volume of the Eridania sea would have been substantial. If you were to normalize these numbers to planetary surface area (which we have not done), the numbers are even more substantial.

2 – Seas are usually saline or chemically rich compared to lakes (especially large lakes). Of course, there is no way to know the salinity of the ancient water body in Eridania, but the presence of chlorides at high elevation imply that it was briny. The lack of sulfates is probably not an indication of a lack of sulfur because there is evidence for sulfides. This means the fluids were probably just chemically rich, but relatively reduced. This makes sense given the independent conclusion that the sea was fed by subsurface flow through mafic-ultramafic crust, rather than overland flow (as most lakes are on Earth).

3 – Lake-floor volcanism and the associated deep water hydrothermal deposits are common on the seafloor on Earth, but *rare in lacustrine environments* on Earth. Because this link is the key element of the story in this paper, it makes more sense to use the term seafloor to avoid confusion. We do not necessarily imply that the seafloor activity means crustal spreading – though it could. It is a fact that these linked basins in Eridania occur in the middle of the strongest remnant magnetism on the planet, which has been interpreted as evidence for crustal spreading (though this is not central to our story).

4 – While we do not know well the nature of early seas on Earth, it seems that they were quite Fe and S-rich, and reduced (euxinic). There is evidence that this was the case in Eridania as well. On Earth, this environment ended due to the gradual oxidation of the surface. Titration of the reduced surface environment on Mars would have occurred much earlier and more quickly than on Earth.

Given all of the differences between typical lacustrine settings and what we see in Eridania, and given all of the similarities between seafloor settings and what we interpret in Eridania, we argue strongly that it would be misleading to use the term “lake” rather than sea.

Reviewer #2 (Remarks to the Author):

This paper focuses on the deep deposits located in the Eridania basin. After showing that their mineralogy is characterized by the presence of saponite, talk-saponite, Fe-rich mica, Fe- Mg- serpentine and Mg-Fe-Ca carbonate, the authors claim that such deposits have formed in deep water, and specifically in an hydrothermal seafloor

setting. Since life on Earth may have flourished in hydrothermal seafloor environments, the Eridania deep deposits would provide an important window not only in the possible past-life search on Mars, but also into the geology of the early Earth.

The paper is clear, it is interesting and provides new insights and interpretation of the discussed topic. Nevertheless, I have few suggestions that are important to be addressed before being published.

Main Text

Line 37: add a footnote or a couple of lines more why Mars is in “many ways” a window into the missing, ancient geological record of Earth.

No one knows what the environment of the early Earth was like, or how early seas formed and whether they were linked to the origin of plate tectonics. But, the earliest evidence for life on Earth corresponds to metamorphosed rocks that seemingly originally formed in a seafloor volcanic-sedimentary environment. The rocks in Eridania are perhaps an unmetamorphosed version of these types of deposits from the same time period.

Line 106 (see comments to the supplementary Images). Please add the errorbars of such measurements.

See below please.

Line 133 Within the Mars discussed spectra, I suggest to add one or two example of these spectra of mixed-layer seafloor clays on Earth.

The reference spectra shown in the diagram include mixed layer clays from Earth (labeled T-S). We have clarified this labeling.

Line 147 I do not get where such deposits are located, please explain that better.

All of the Fe-rich samples are labeled on the mineralogy overview map, which is now Figure 5.

Line 185. Why 10%? Please provide a citation for this number.

Added a reference to support the relationship between crater diameter and exhumation depth.

Line 289. Reference 48 is ok for Mars, but what “other planets” do you mean? Please provide examples.

We removed this text, but we were referring to lunar mare.

Line 291-292. You state “The most plausible way to produce such large volumes of deep basin, deep water deposits is through seafloor volcanic-sedimentary processes.” So you mean hydrothermalism on Mars. How do such deep basin deposits presented in Figure 3 compare with the example of mounds published by Marzo et al., 2010 (Evidence for Hesperian impact- induced hydrothermalism on Mars, Icarus 208, 667-683, Fig. 8)? Are the shapes comparable?

The materials within the Eridania basins are fundamentally different from those observed in the central peaks of martian craters, including those published by Marzo. In terms of geologic context, the similarities are that they both occur in basins (though large basins in this case and smaller ones in the case of altered central uplifts), and the differences are many. Central peaks contain fractured uplifted material that, in the case of Marzo’s work, is argued to have been overprinted by younger hydrothermal activity. In other words, the hydrothermal deposits would occur as “pasted on” or zoned materials on or within the fractured central uplift. In this case, we are talking about massive, but horizontally extensive materials that are intensely altered. In terms of the mineralogy, the spectral shapes might be similar in the sense that they both show Fe-rich clays, but we have not done a detailed analysis of the clays in that paper.

This comment is just out of curiosity: Is there any CRISM observation of the Gorgonum deep deposits? Is there not even a single observation performed by CRISM or OMEGA supporting the paper’s interpretation?

We have clarified this issue in the manuscript as it is a fairly important point. There are indeed many CRISM observations in Gorgonum, but none of them show clays. In any case, the morphology and context of these units indicates clearly that they are not “deep basin deposits” as are observed in the other basins. We mentioned in the previous version that the deposits in the eastern basins are spectrally unremarkable. But we have emphasized that point here.

Supplementary Images

Figure S1.

Explain what $\Delta Br/\Delta Lat$ is. What is K concentration? In the caption, 106 km² should be 10⁶ km²

Fixed

Figure S2.

Add the errors of the model ages presented. Write down what is the dimension range you are using to compute the model age. The crater retention ages computed with craterstats 2 are model ages. Change the word “ages” or “surface ages” with

“model ages” throughout the entire text.

We understand where the reviewer is coming from here, but we prefer not to change the text to read “model” age throughout. We have changed the term to “model age” in the initial report of the results within the text. We wish to keep the text as direct as possible considering the wide audience. Most readers should understand that the ages are in fact modeled. To be clear, though they are totally different, all radiometric ages are also model ages.

Properly explain how you have selected the two areas inside the Gorgonum and Ariadnes. Is it an elevation criterion? A morphological criterion? If so, show the morphological map in the supplementary material. Moreover inside the Gorgonum area, explain why there is a hole in the middle where craters have not been computed.

We have chosen the polygon in which to count craters according to geomorphologic and geologic mapping & interpretation. The polygons delineate the extent of the “knobby terrains” corresponding to “colles” or “chaos” in each case. The “hole” in the middle of Gorgonum corresponds to the outline of a totally different geologic unit. It wouldn’t make sense to count craters on two different geologic units and lump them together.

Figure S3.

Caption: ..of an unnamed *basin*? Do you mean basin, correct?☒

We have deleted this figure.

How do you know that the volcanics are Hesperian? Please add a reference for that.

We have deleted this figure.

Add scalebars for a), b) and c).☒

We have deleted this figure.

Add a zoom to properly see the limit between the Noachian deep basin deposits and the Hesperian volcanics.

We have deleted this figure.

Figure S4.

Add a scalebar to b)☒

We have deleted this figure.

Plot d) what is the difference between the Ankerite red plot and the Ankerite gold plot? Why are they different? Is the grain size different?

We have deleted this figure.

Figure S5.

Add a scalebar to a) and b)☒

We have deleted this figure.

Show the Regions of Interest (ROI) where you have extracted the spectra.☒

We have deleted this figure.

How many pixels are you evaluating to get the presented spectra. Add that number in the caption.

We have deleted this figure.

Figure S6.

Show a map on top of the four considered subbasins, with a line indicating where you have extracted the profiles otherwise I do not have the possibility to understand where you have extracted them.☒What are the differences in ages of the Eridania and the for below basins?

We prefer not to add figures showing the profiles for each because this just adds more data and distracts from the focus of the paper. We are citing a relationship that has already been established. No matter which direction one draws a profile through each basin, the highlighted relationships hold true.

Supplementary materials

Line 41: add a reference to support this sentence.

OK

Line 76 Mars not mars

Fixed

Reviewer #3 (Remarks to the Author):

General comments:

This paper describes the geology and the mineralogy of Eridania Basin using the data acquired by CRISM, HiRISE and CTX instruments. Purpose of the manuscript is to demonstrate that deposits in Eridania formed in a hydrothermal seafloor setting. Although I'm not a geologist, I believe that the paragraph about the geology is described adequately and I appreciated the general discussion done. Regard to the mineralogical discussion, most of the information needed to understand the presence of various mineral are given, but in some cases (e.g. jarosite) the presence of the mineral is not yet sufficiently proved. For this reason the author should at least address the points listed below. Overall, the text is clear and accessible to nonspecialist. If these points will be clarify my recommendation is for acceptance.

Specific comments are found below. Review by F.G. Carozzo

Line 58. In figure 1, it is not clear where is Eridania Basin. In the figure are present various regions such as Gorgonum Chaos and I do not understand if these areas are located in Eridania Basin. The author might consider to draw the boundaries of Eridania using a curve.

We have clarified the text to indicate that the "Eridania basin" is in fact the suite of sub-basins including Gorgonum, Atlantis, etc. There is a new Figure 1 that explains this issue.

Lines 106. The author cites supplementary materials referring to the Figure S-2. In the caption remove "for are shown" and add "are shown" after "Gournum (d)".

OK

Line 113. Where the kipukas are located? I suggest to add a reference to a figure or to add a new figure. The term is specialistic and a figure can help the reader to understand its nature.

We have clarified this in the caption

Line 128. Talc has a set of absorptions (1.44, 1.53, 2.0-2.18) not present in the CRISM spectra of figure 4. It is useful to show an example of talc spectrum. Why the author cited the talc? I believe that the yellow spectrum of Figure 4a is saponite. Can the author explain where the T-S library spectra have been taken? What are the samples?

We have updated the text to indicate that the talc and talc-saponite mixed layer samples are from seafloor environments on Earth.

Talc is a complicated mineral because the most familiar form for many geologists is

very well ordered, very Mg-rich talc that tends to have a number of sharp infrared features, including those identified by the reviewer. We have found, however that the substitution of Fe into talc results in decrease in the ordering as measured by infrared. In other words, some of these features pointed out by the reviewer are not present in talc from seafloor environments. In fact, there is some mixing of terms in the literature regarding talc, saponite, and talc-saponite. Both talc and saponite are TOT clays meaning that they are tetrahedral-octahedral-tetrahedral clays with basal spacings measured in XRD at nearly 10 Å. However, talc is slightly lower and is not expandable due to hydration and saponite is higher than 10 Å due to expansion. These two phases are often interlayered at the lattice scale, which means that they can only be characterized through detailed work (which is often not carried out completely).

In this paper, we simply wish to point out that in our work on seafloor hydrothermal clays on Earth, we found talc-saponite mixed layer clays and that the infrared features of these clays are similar to those we have identified on Mars.

Line 143. The author cited mica as possible candidate due to the absorption at 2.295-2.305 μm . Micas showing this band (e.g. glauconite) have other absorptions at 1.91 μm and at 2.36-2.37 μm that are not present in the spectra of figure 1a. How does the author explain it?

We appreciate the reviewer's comment and we acknowledge that there is not a perfect match between the measured data and Fe-rich mica such as glauconite. However, we do find clays that are demonstrably Fe-rich based on the position of the metal-OH absorption at <2.305 microns in spectra that lack strong water absorptions at 1.9 microns. So, these are Fe-rich TOT clays that are not expandable. They could be Fe-rich mica, but not actually pure glauconite. Furthermore, these materials on Earth (including in seafloor settings) are often mixed-layered with smectites, further complicating matters. The main point is that smectite clays alone will not explain the observations, and Fe-rich illite is a good candidate in some cases.

Line 147. Other minerals have the same doublet (e.g. mica or amphibole). Why these minerals are excluded?

We mention that mica is a good candidate. Amphibole might exist, but we have yet to find an observation that provides an adequate match to amphiboles. One complicating factor is that amphiboles are extremely well crystalline compared to clays and their very sharp, deep features are not observed in the data from Mars.

Line 159. The author cited diagnostic features useful to detect jarosite. From the figure 4b it is difficult to distinguish the position of the minima. May the author show me a comparison of the main bands using a close up (between 1.8 and 2.5 μm)? In my opinion the band at 2.46 is not present in the CRISM spectrum.

In the same figure it is present a line near 2.3 μm showing the presence of the spectral feature in both CRISM and library spectra, but why the band at 2.21 μm is not discussed? In my opinion, sentence at lines 159-161 does not justify the absorption because it is at longer wavelengths. In the CRISM spectrum the band at 1.85-1.9 μm , diagnostic of jarosite, is not present. It is after 1.9 μm . Can the author explain better why there is this difference?

Jarosite is one of the most complicated minerals to analyse using infrared because it has many spectral bands (which is good), but they shift around and sometimes disappear (which adds complication). Even the main papers using CRISM to identify jarosite on Mars (Milliken et al., *Geology*, 2008 and Farrand et al., *Icarus*, 2009) do not provide perfect matches. These mismatches are understandable given the complex relationships between jarosite crystal chemistry and infrared features (Bishop et al., *American Min.*, 2005). Based on our experience, there is definitely jarosite in these deposits, though we acknowledge that none of the jarosite detections on Mars is a perfect crystal chemical match to those samples in our libraries.

Line 170. Figure 4D is not present.

Updated.

Paragraph Carbonates, Line 167. The CRISM spectrum a3 (in figure 4b) has a broad absorption near 1.9 μm that is not present in the library spectrum. This band and the other minima can be consistent with the presence of hydrated phases, for example a mixture of carbonates (e.g. calcite, magnesite) with Mg-phyllsilicates (e.g., vermiculite, saponite). The author should discuss it as suggested for example in Carrozzo et al. 2017 (*Icarus*, "Geology and mineralogy of the Auki Crater, Tyrrhena Terra, Mars a possible post impact-induced hydrothermal system").

We acknowledge that the hydration in the carbonate on Mars might be due to mixtures with hydrated silicates.

Line 174. Why the author show the spectrum of ankerite? The minima of absorptions of spectrum a3 are different

We have shown it for the sake of comparison. In the case of carbonates, the two main C-O absorptions shift position as a function of cation chemistry. We do not have any exact matches in terms of band position (chemistry) to the martian carbonates. But, the position and relative strengths of the bands in ankerite provide insight to help interpret the IR spectra

Paragraph Chlorides, Lines 198-208. The discussion about the absorptions of chlorides is not present. Moreover, in figure 4 the spectra are not present.

It is stated in the text that the identification of chlorides is based on thermal IR data,

and the detections are referenced from the literature, not from our detections. Because they are detected at different wavelengths by different authors, we do not show them on our plots. Actually, they do not have NIR features in any case – only a slope.

Lines 222-225. I'm not sure that the mineral spectra cited are present in figure 4. If not present, the comparison of these CRISM spectra with library ones is useful.

One of the main points of this paper is that authors have previously mixed interpretations among the various basin deposits, including the very old deep basin deposits and the younger volcanic resurfacing. We highlight how the volcanic resurfacing is different; it is this material that contains the pedogenic material. So, further description of these deposits distracts from the point of this paper. For this reason, we have chosen not to confuse the issue at hand (deep basin alteration) by adding these materials.

Lines 232-236. This is a crucial point. This is well described thanks to the figure 5A but not to the figure 5C. Add a reference to the figure 5.

We have clarified a reference to the Figure on chlorides, which is now called Figure 8.

Line 251. The author writes “chlorides are present at higher elevations along the interior basin margins”. The spatial distribution seems to contradict this sentence. In figure 1, yellow points are located inside Aridnes Colles and Atlantis Chaos.

We have tried to clarify this issue using map symbols of different shapes.

Lines 253-255. The author affirms that deposits where chlorides are found do not include any sulfates, but in figure 1 they seem to be located in the same areas.

In that particular area, the two deposits are located together. But as you see in our chloride figure, the chlorides underlie the volcanic resurfacing and it is within the volcanic resurfacing that the Al-clays and Ca-sulfates occur. So, their co-location in horizontal space is not co-occurrence in terms of environment. In other words, the sulfates are pedogenic.

Lines 310-314. A short discussion should include hydrated silica and sulfates. In particular I'm not understanding the importance of sulfates in a hydrothermal system. In the paragraph about the jarosite (see lines 155-165) the author writes that this mineral occurs with clay. How does the author explain the occurrence of both minerals in the same place?

One of the interesting things we found in this work is that jarosite is observed in deposits where it should not be stable. In other words, it is found in the exposed walls of blocks of material containing Mg-rich clays. These two types of phases

should not be deposited together. But they occur together here because the jarosite was not actually deposited as jarosite, but as Fe-sulfide. That sulfide was later weathered to jarosite upon exposure to the martian atmosphere.

Figure 1. It is difficult to distinguish jarosite from chlorides and Fe-clays from carbonate in the map. Please, change the colors or symbols.

Hopefully the updated map is clearer.

Figure 4. What is the meaning of T-S? Maybe Talc-Saponite? Can you explain it in the caption? In the figure the author shows the ratioed spectra. I would like to see the original spectra and the various spectra of neutral components used to obtain the ratioed spectra. Where the “neutral” (or “featureless”) spectra are taken? This should be present in the supplementary materials. In Figure 4a is not clear the match between the single CRISM spectrum and the library one.

We choose not to include all of the reference spectra and neutral surfaces because it is distracting for people who are not used to looking at spectra every day. Ten years ago when the first CRISM papers were being published, it was very important to show the reference spectra because we were all trying to find our way. But now, there are many papers published on the subject and the methods of spectral transformation are straightforward. There is no added information from including the data. So, we prefer not to include them in order to clarify issues for the reader.

Figure 5C. Chlorides are only found at high elevations but in figure 5C this point is not clear. Where the clays are located? I suggest to include another figure where the elevation is present because at the moment the perception is that chlorides occur at lower elevations.

Please note that the locations of chlorides are improved on the mineralogy summary figure now (previously Figure 1 and now Figure 5). With this map, one can see that the chlorides occur at high elevations. This is again shown quantitatively in figure 8.